# Drivers Underlying Metastasis and Relapse in Medulloblastoma and Targeting Strategies

**DOI:** 10.3390/cancers16091752

**Published:** 2024-04-30

**Authors:** Karl O. Holmberg, Anna Borgenvik, Miao Zhao, Géraldine Giraud, Fredrik J. Swartling

**Affiliations:** 1Department of Immunology, Genetics and Pathology, Science for Life Laboratory, Rudbeck Laboratory, Uppsala University, 75185 Uppsala, Sweden; karl.holmberg_olausson@igp.uu.se (K.O.H.); miao.zhao@igp.uu.se (M.Z.); geraldine.giraud@igp.uu.se (G.G.); 2Department of Pediatric Oncology, Dana-Farber Cancer Institute, Boston, MA 02215, USA; annam_borgenvik@dfci.harvard.edu; 3Harvard Medical School, Boston, MA 02115, USA; 4Broad Institute of MIT and Harvard, Cambridge, MA 02142, USA; 5Department of Women and Child Health, Uppsala University, 75124 Uppsala, Sweden; 6Department of Pediatric Hematology and Oncology, Uppsala University Children’s Hospital, 75185 Uppsala, Sweden

**Keywords:** pediatric brain cancer, leptomeningeal spread, SHH, MYC, relapse, metastasis, targeted therapy

## Abstract

**Simple Summary:**

In this review, we summarize reported molecular mechanisms underlying tumor progression and relapse of medulloblastoma, one of the most frequent malignant pediatric brain tumor entities. Medulloblastoma relapses are difficult to treat, and patients have, overall, a poor prognosis. Apart from describing the biology promoting brain tumor spread, the review will also highlight important preclinical models used to study leptomeningeal disease and recurrence. Finally, we identified clinical trials for medulloblastoma relapse and will discuss novel attempts to target therapy-escaping cancer cells responsible for recurrence.

**Abstract:**

Medulloblastomas comprise a molecularly diverse set of malignant pediatric brain tumors in which patients are stratified according to different prognostic risk groups that span from very good to very poor. Metastasis at diagnosis is most often a marker of poor prognosis and the relapse incidence is higher in these children. Medulloblastoma relapse is almost always fatal and recurring cells have, apart from resistance to standard of care, acquired genetic and epigenetic changes that correlate with an increased dormancy state, cell state reprogramming and immune escape. Here, we review means to carefully study metastasis and relapse in preclinical models, in light of recently described molecular subgroups. We will exemplify how therapy resistance develops at the cellular level, in a specific niche or from therapy-induced secondary mutations. We further describe underlying molecular mechanisms on how tumors acquire the ability to promote leptomeningeal dissemination and discuss how they can establish therapy-resistant cell clones. Finally, we describe some of the ongoing clinical trials of high-risk medulloblastoma and suggest or discuss more individualized treatments that could be of benefit to specific subgroups.

## 1. Introduction

Medulloblastomas (MBs) are malignant pediatric brain tumors currently classified into the following four molecular subgroups: WNT, SHH, Group 3 and Group 4 [1]. They arise in the developing cerebellum throughout childhood, with an incidence of 0.5 cases per 100,000 people and with a median diagnostic age of 6 years [2].

MB treatment is multimodal with surgery, followed by craniospinal radiation therapy and chemotherapy. Five-year overall survival (OS) in high-income countries for MB exceeds 80% for standard-risk (SR) (“average risk” in US) and is approximately 60% for high-risk (HR) patients [3]. Infants and very young children (under 3 years of age) with MB have a worse 5-year progression-free survival (PFS) and OS compared to older children [4,5,6,7]. Radiotherapy administered to the developing brain and spine has been associated with severe growth side effects [8,9,10] and unacceptable detrimental neurocognitive deficits, hampering the independence and quality of life of these young children [11,12,13]. The current SIOPE HR MB trial [14] proposes an alternative arm with high-dose chemotherapy (HDCT); likewise, the actual infant strategy is using HDCT with the aim to improve outcomes and decrease or avoid the use of irradiation. The different arms, resulting in higher survival rates [15,16,17,18] but curing only around half of the patients, are comparing high-dose chemotherapy, hyperfractionated–accelerated radiation (HART arm), and conventional therapy (e.g., 36 Gy), followed by maintenance treatment. Despite these multimodal treatment strategies, about 15% of SR, and 23% of HR, patients develop a relapse.

Most relapsed medulloblastomas come with an extremely poor prognosis. There is currently no international consensus on treatment, but these patients are often enrolled in clinical trials. There is a pressing need to identify MB patients with an increased risk of developing a relapse and test new treatments to avoid recurrence. What is more, the four subgroups present distinct clinical characteristics, including heterogeneous recurrence patterns and the propensity for metastatic dissemination.

### Metastatic Disease and Relapse Patterns in MB Subgroups

In 20–30% of affected children with MB, metastasis is evident at the time of disease presentation, while in others, metastases are shown first at recurrence [19]. Within the main MB subgroups, SHH and Group 4 patients have an intermediate prognosis (67–88% 5-year OS [20,21] and can present with primary metastatic dissemination (approx. 15–20% and 40–45%, respectively) [21]. Group 4 tumors, being the largest subgroup, also comprise the majority of MB cases with primary metastasis. WNT subgroup tumors rarely metastasize (10–12%) [21], and even when they do, the disease prognosis is excellent [22]. By contrast, Group 3 tumors are associated with the poorest outcome (41–66% [20,21]) and show the highest frequencies of metastatic dissemination (>55%) [21] at first presentation, especially those harboring *MYC* amplifications [23,24] (Figure 1). SHH-activated tumors with *TP53* mutations also have a very poor prognosis and show a higher degree of leptomeningeal spread compared to their *TP53* wild-type counterparts [3,20]. That said, the SHH β MB subtype (67% 5-year OS) has an even higher frequency of metastasis at diagnosis as compared to the SHH α MB subtype (70% 5-year OS) that harbors the most *TP53* mutated cases [21].

The presence of metastasis or leptomeningeal disease (LMD) is markedly higher in MB relapse as compared to primary diagnosis. This suggests that the process of metastasis is tightly connected to tumor progression and relapse, at least in non-WNT MBs. It is expected that 70% of relapsed MBs have LMD [19], which is likely an underestimation explained by the lack of standardized protocols for screening microscopic disease.

Relapse incidence has been reported to depend on patient age. Younger patients (0–5 years) relapse to a higher extent than children and adolescents (older than 3 years) [25]. This is most likely due to the lack of viable treatment options for infants and young children who are ineligible for craniospinal irradiation (CSI). Others report MB recurrence rates of around 30% overall for subgroups and patients [26]. The relapsing tumor usually stays within the same subgroup as the primary tumor [24,27]. While SHH MBs show significantly more local relapses in the area where they originally arose (cerebellum), Group 3 and 4 MBs usually present with distant relapses or leptomeningeal spread [19]. Metastatic relapses of Group 4 MBs tend to present with isolated deposits, whereas the metastatic relapses of Group 3 MBs are typically multifocal or laminar.

In about 5–7% of all MBs, high-grade gliomas arise as secondary malignancies [25,26]. These are believed to be irradiation-induced secondary tumors rather than relapses, as such [28]. About 9% of long-term survivors of primary CNS cancers will have developed a second CNS cancer 40 years later [29]. Irradiated pediatric MB patients have a higher risk of developing a secondary brain tumor (mostly meningiomas but also glioblastomas) as compared to other brain tumors, excluding pediatric patients with glioblastomas. The rate of secondary malignancies might be lower after proton beam therapy as compared to photon therapy [30], as proton therapy will give lower radiation doses to the surrounding normal brain. Large initiatives to study secondary malignancies coupled to types of radiation treatments in pediatric patients are currently ongoing [31].

When studying the overall 5-year mortality among all subgroups, the majority of patients that eventually die from a MB were first diagnosed as a Group 3 or a Group 4. In a large MB cohort of patients older than 3 years, Group 3 MBs show the fastest time to relapse (0.9 years) closely followed by SHH MBs (1.2 years), with Group 4 MBs presenting a rather long (2.74 years) median time to relapse [25] similar to that observed in WNT MBs (2.77 years) [22] (Figure 1).

**Figure 1 cancers-16-01752-f001:**
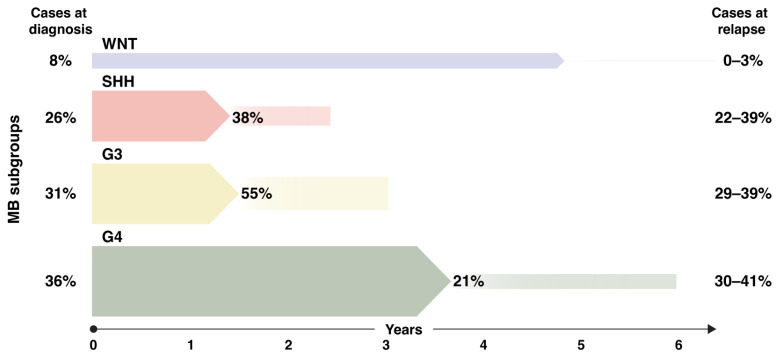
Relapse frequency depending on medulloblastoma subgroup. Cases at diagnosis represent the percentage of a particular medulloblastoma (MB) subgroup as compared to all MBs. Percentages within the figure represent the relapse fraction for each subgroup. The initial arrow length indicates the time to relapse from diagnosis in years and the smaller arrows with sharp ends represent the time from relapse to death. Data were obtained from [24,25,32] and do not include secondary non-MB relapses. WNT: WNT-activated, SHH: Sonic hedgehog activated, G3: non-WNT/non-SHH Group 3. G4: non-WNT/non-SHH Group 4.

## 2. Biology of Medulloblastoma Metastasis and Relapse

### 2.1. Leptomeningeal Disease

LMD is the detection of cancer cells within the meninges (arachnoid and pia mater) or in the subarachnoid space filled with cerebrospinal fluid (CSF) that continues down the spinal cord. LMD is found in 1–8% of all cancer patients and in 10–20% of primary CNS tumors [33]. Compared to adult brain tumors, malignant pediatric brain tumors more commonly disseminate (20–30%). During LMD, it is believed that metastatic cells have left the primary tumor site, infiltrated the meninges, and spread via the arachnoid space or the CSF to form a metastatic colony. As MB cells have also been discovered in the bloodstream of patients, LMD might arise due to the spread of circulating tumor cells through the blood [34].

What are the mechanisms behind MB metastases and disseminated disease? One could speculate that there is a constant release of tumor cells into the CSF from primary or macro-metastatic tumor sites or that circulating tumor cells are different than the cells in the bulk of the tumor, giving them a survival advantage in the CSF. The CSF is low in nutrients compared to the environment at the primary location in the brain. The total protein concentration in the CSF is 50–100 times lower as compared to plasma [35]. Moreover, the CSF, with its constant and turbulent flow, is an environment in which not all types of tumor cells can thrive. The molecular process in which MB cells acquire abilities to cause LMD likely involves genetic or epigenetic changes. Epithelial-to-mesenchymal transition (EMT) is a process characterized by the suppression of epithelial markers, upregulation of mesenchymal markers, and the loss of cell adhesion. EMT promotes metastatic spread in other types of tumors (e.g., breast cancer [36]) and facilitates their spread to the CSF and the central nervous system (CNS). Molecular changes or mutations driving EMT-like transition in MB will provide tumor cells with the capability to invade meningeal or endothelial barriers [37], but also likely protect the tumor cells from cell death. Eventually, these changes could make them dormant and thereby resilient to radiation and chemotherapies targeting rapidly dividing cells. Dormant cells might develop a more efficient therapy resistance with time if they persevere. With acquired resistance, or upon treatment cessation, dormant tumor cells could start dividing and expand their growth. In many cases, mesenchymal cancer cells that disseminated and formed new colonies concomitantly revert to an epithelial phenotype in a mesenchymal-to-epithelial transition (MET). It has been suggested that this could happen also in MB [38]. If this is the case, targeting the awakening of dormant cells or a MET state would perhaps be a useful strategy in relapsed disease. With this in mind, there are efforts in breast cancer research aiming at targeting tumor dormancy that could be useful also for the treatment of MB relapse. For example, Aouad et al. suggest that stabilizing a MET cell state and inhibiting the block of dormancy exit could be critical to preventing breast cancer recurrence [39]. We will discuss such treatment ideas and other inhibitory strategies in more detail below.

### 2.2. Means to Follow Metastatic Cells and Tumor Relapse

Therapy intensification might enable salvage of patients with persistent disease. Better detection of microscopic disease would lower the threshold of minimal residual disease (MRD) and complement staging after complete surgical resection. As mentioned earlier, no molecular biomarkers are currently used to monitor therapy response in brain and spinal cord tumors in children [40]. Thus, the potential persistence of microscopic disease is not detected, and this delays the relapse diagnosis until it is macroscopic and often widely disseminated. Identifying patients with excellent therapy responses and the absence of MRD may enable treatment de-escalation to reduce the risk of long-term detrimental effects. Recently, whole genome sequencing (WGS) on CSF was reported to rather successfully follow MRD in MB [41]. Apart from WGS, cell-free DNA from the tumor (ctDNA) has emerged as a potential biomarker for diagnosis, prognosis, and MRD detection [42]. Analysis of ctDNA in CSF from brain tumor patients was first reported in 2015 [43] and patient-specific multiplex ddPCR on CSF from MB can potentially be used for relapse monitoring [44].

Analysis of proteins or peptides from CSF has been shown to identify disease-specific markers with the potential to track residual disease and relapse [45] in different medulloblastoma subgroups present at different localizations [41,46]. However, tumors in MB patients that do not present with LMD at primary diagnosis also have detectable ctDNA, especially if the original tumor is situated within or next to, e.g., the 4th ventricle [44]. Thus, although ctDNA can identify MRD, it may not serve to measure LMD rates in these cases. Here, serial cytology analysis of CSF to detect metastatic disease is still the gold standard. For LMD detection, magnetic resonance imaging (MRI) of the neuroaxis can be useful, and even more so if the technique is further improved [47].

## 3. Molecular Signatures of Medulloblastoma Relapse

Several clinical investigations, including extensive molecular profiling, serve as good starting points regarding the pathways associated with recurrent MB [24,25,26]. Still, although certain mutations or aberrant signaling pathways are accumulated in distinct subgroups, there is no consensus on what aberrations would be considered signature markers of MB relapse. Some studies highlight a correlation between molecular profiles and distinct signaling pathways involved in the dissemination of MB metastases [48]. Conclusions are often derived from the observed upregulation of “pre-metastatic” genes that are therefore suggested to be the genetic drivers of tumor cell spread [49,50]. It is clear, from studies of matched sets of primary-metastatic biopsies using expression and methylation profiling [51], that the molecular subgroup is stable between primary and metastatic disease. Together with recent single-cell sequencing data [52,53], this suggests that the MB subgroups arise from distinct cells of origin but also that the primary and metastatic compartments of an MB are sharing the same cell of origin. Another strategy to study tumor cell spread and relapse involves developing animal models using clinically relevant drivers of MB subgroups, but by also adding distinct genetic modifications that would promote tumor progression and dissemination to study the metastatic process. Below, we will present some of these efforts and discuss a selection of relapse mechanisms reported in the various MB subgroups.

### 3.1. WNT Medulloblastoma

WNT MBs rarely relapse, and if they do, they mainly recur because of inferior treatment and often present as metastatic disease located in the lateral ventricles [22]. WNT tumor relapses have been associated with enrichment for *TP53* mutations [54] (Figure 2). Contrary to other subgroups, metastases in WNT tumor relapses do not seem to confer a worse prognosis [55]. Perhaps this is explained by the reported leakiness of the blood–brain barrier and associated enhanced standard drug perfusion of WNT tumors [56].

Some MB subgroups commonly display *MYC* or *MYCN* amplifications with concomitant p53 loss at relapse. Compared to Group 3 MB, WNT MBs rarely present with *MYC* amplifications but are enriched in gene sets related to extracellular matrix (ECM) regulation and cellular adhesion. The expression of mutationally-stabilized *Myc^T58A^*, in a murine WNT MB model (Table 1), significantly accelerated growth, resulting in metastatic disease and disruption of ECM and cellular adhesion pathways [57]. Achiha et al. studied the expression of a transmembrane glycoprotein, cluster of differentiation 166 (CD166), or activated leukocyte cell adhesion molecule (ALCAM) in all MB molecular subtypes using humane MB samples, documenting a major expression in WNT MBs. Moreover, they reported that ALCAM in vitro appeared to be involved as a positive regulator of the proliferation and migration of MB tumor cells, whereas ALCAM silencing in vivo enhanced tumor cell invasion at the dissemination sites [58].

### 3.2. SHH Medulloblastoma

Mutations causing increased activity of the Shh pathway have been identified in up to 20% of MB cases. These include inactivating mutations of *PTCH1* or *SUFU*, genes that encode negative regulators of Shh signaling, or activating mutations of *SMO*, a gene that encodes the Shh mediator, Smoothened. In the SHH subgroup, patients with somatic *TP53* mutations are stratified as high risk, with a worse prognosis and the highest percentage of relapse [59]. There is also a co-occurrence with *MYCN* amplification and chromothripsis further linked to relapse [26,59,60]. A comparison of infant and non-infant SHH MB highlights the higher proportion of chromosomal arm-level CNV changes in non-infant cases but more putative driver mutations in SHH MB relapse in infants [54]. The large-scale chromosomal changes in the non-infant cases are likely due to the use of CSI. As previously mentioned, SHH MB mainly recurs locally, suggesting that molecular features specifically tied to metastatic spread and relapse might be limited [24].

Comparative studies of the transcriptome of matched primary-relapsed MB samples showed the impact of age where changes in gene expression profiles were more pronounced in the younger SHH-MB patients with relapses, likely due to increased numbers of DNA mutations or aberrations; relapsed SHH tumors also became less differentiated [27]. In the minireview by Li et al. [50], published before the latest WHO CNS5 classification, several associations between determined MB molecular profiles and their capacity for LMD were reported. Several essential signaling pathways were implicated in LMD: in particular, a pro-migration signaling axis consisting of the platelet-derived growth factor receptor (PDGFR), SRC, GRK6, and CXCR4, which were found in SHH MB cells along with the expression of c-MET, a promoter of tumor cell dissemination. The PDGF receptor expression has previously been associated with MB relapse [61]. Here, PDGF receptor beta rather than alpha were found to be upregulated in samples from patients with metastatic MB [62]. Further, MB cell migration was shown to be dependent on the PDGF-driven ERK-mediated activation of p21-activated kinase 1 (PAK1) [63].

We recently showed that MYCN, through activation of OCT4, drives mTOR signaling in human neuroepithelial stem cell/iPS-derived stem cell models of SHH MB. Increased levels of OCT4 led to increased mTOR signaling and metastasis [64]. Dual PI3K/mTOR inhibition prolonged survival and reduced LMD in vivo. In line with clinical data and other studies, reduced p53 activity (introduction of dominant-negative p53 mutant) also promoted tumor formation and LMD in our avatar models. Further, AMP-activated kinase (AMPK) inactivation slows the progression of SHH-driven medulloblastoma. Disabling AMPK in reduces mTORC1 activity and impairs MB stem cell (OLIG2-positive) populations [65]. AMPK can further suppress metastasis of SHH-MB cells by restraining the activation of SHH and NF-κB pathways where the dual blockade of these pathways has a synergistically therapeutic effect on SHH MB [66]. In SHH MBs, the concept of the cell state as a player in relapse formation with quiescent SOX2-positive cells is suggested to drive growth and relapse in SHH MB [67] (Figure 2).

Since the development of the first *Ptch1*+/− GEM model in 1997, a plethora of SHH MB mouse models (Table 1) have been developed using multiple approaches [68]. The transcription factor ATOH1, a key regulator of cerebellar development, plays a crucial role in SHH MB development, metastasis, and recurrence. Deletion of Atoh1 prevented cerebellar neoplasia in an MB mouse model driven by an activated *SmoM2* [69] mutant previously identified in human basal cell carcinomas. The overexpression of Atoh1 (aka. Math1) in *Ptch1*+/− mice led to highly penetrant MB development at a young age with extensive leptomeningeal disease and metastasis to the spinal cord and brain. Metastatic tumors retained abnormal SHH signaling as tumor xenografts. Conversely, ATOH1 expression was detected consistently in recurrent and metastatic SHH MB. Among these targets specific to metastatic tumors, there was an enrichment in those implicated in extracellular matrix remodeling activity, the cytoskeletal network, and interaction with the microenvironment, indicating a shift in transcriptomic and epigenomic landscapes during metastasis [70]. Recently, the Sleeping Beauty (SB) transposon system was shown to be an effective tool for functional genomics studies of solid tumor initiation, progression, and metastasis [48] When these mice were bred with transgenics for a concatemer of the T2/Onc transposon in a *Ptch1*+/− background, the *Math1*-SB11/T2Onc mice showed increased penetrance (39% to 97%) of MBs as well as decreased tumor latency (8 months to 2.5 months). Although *Ptch1*+/− MBs are usually localized, the addition of SB transposition results in metastatic dissemination with drop metastases through the cerebrospinal fluid pathways [48]. *Eras*, *Lhx1*, *Ccrk*, and *Akt* belonged to candidate metastasis genes found in or around common insertion sites, in which transposon insertions occurred significantly more frequently than expected by chance. Ectopic expression of any of these genes in Nestin+ neural progenitors in the cerebella of mice, by retroviral (RCAS) infection in combination with *Shh*, promoted significant dissemination and leptomeningeal tumor spread in the spinal cord [71] (Table 1).

The overexpression of SmoA1 with a constitutively active point mutation from the *NeuroD2* (ND2) promoter induces a 48% incidence of MB formation [72]. Tumors form in 94% of homozygous Smo/Smo mice by 2 months of age [73]. The *SmoA1* GEMM faithfully recapitulates aggressive human SHH MB, with metastases observed in ~30% of mice. Eight to fifteen percent of SHH-MB patients have somatic mono-allelic mutations in the chromatin modifier and methyltransferase *KMT2D* that are predicted to result in a truncated protein. Using two sporadic SHH genetic models, Smo activation (SmoM2) and Ptch1 loss (*Ptch1*fl/fl) combined with N- and C-terminal floxed truncating alleles of *Kmt2d* showed that heterozygous or homozygous loss of *Kmt2d* greatly accelerates SHH-MB penetrance and dissemination compared to mice with only an SHH pathway-activating mutation. Most notably, heterozygous loss of *Kmt2d* is sufficient to drive fully penetrant leptomeningeal metastasis to the spinal cord, a hallmark of advanced-stage disease. Tumors lacking Kmt2d have upregulation of the genes implicated in tumor metastasis including cell migration and EMT [74] (Table 1).

*TP53*, as a constitutional mutation, defines the Li–Fraumeni syndrome. These patients have an increased risk of developing MB. Although no MBs were found in mice with mutant *Trp53*, 40% of *Trp53mut*/Math1-SB11/T2Onc mice developed disseminated MB [48]. Like human MBs with *TP53* mutations, *Trp53mut*/Math1-SB11/T2Onc MBs present with large cell/anaplastic histology (Table 1).

### 3.3. Group 3 Medulloblastoma

Tumors in high-risk Group 3 MB patients often carry *MYC* amplifications linked to metastatic recurrence [24] and predict a poor outcome [75,76,77]. Relapsed Group 3 MBs present with a limited number of new mutations, potentially Chr 2q gain and Chr 15 loss [54], suggesting that the molecular state of primary disease is already prone to metastasis and relapse.

In 2010, our group generated the first Group 3-like transgenic mouse model by overexpressing MYCN from the Glutamate transporter 1 (Glt1) promoter [78]. Somatic *Trp53* DNA-binding domain mutations are found in a large proportion of GTML tumors. Mice completely deficient in p53 (GTML/*Trp53*^KI/KI^) developed tumors with dramatically increased penetrance. In vitro, treatment of GTML/*Trp53*^KI/KI^ tumorspheres with the Aurora-A kinase inhibitor (Alisertib) destabilized MYCN via disruption of the Aurora-A/MYCN complex and caused growth inhibition comparable to doxycycline-mediated suppression of MYCN expression [79]. This highlights a central role of key transcription factors in the regulation of cell plasticity in Group 3 MB recurrence. We identified that SOX9-positive cells are accumulating in paired samples of primary:relapsed Group 3 biopsies and we functionally illustrated their essential importance in relapse formation in our model resembling MYC-driven Group 3 MB. SOX9 expression promoted EMT, and LMD and led to cells entering a dormancy state to cope with treatment or MYC depletion [80] (Table 1).

In Group 3 tumors, *MYC* amplifications are at least 3–4 times as common as *MYCN* amplifications [81]. To clarify the extent of MYC involvement in the pathogenesis of MB, we crossed TRE-MYC mice with a tTA-expressing strain under the control of the Glt1 promoter. Resultant brain tumors developed sporadically with an average latency of 90–150 days and about 60% penetrance. Compared to MYCN-expressing brain tumors driven from the same promoter, pronounced ARF silencing is present in our MYC-expressing model and in human MB. While Arf suppression causes increased malignancy and increased photoreceptor-negative high-grade glioma formation in MYCN-expressing tumors, it caused increased malignancy with significantly elevated LMD in photo-receptor-positive MYC-expressing tumors [82] (Table 1).

Another glial transcription factor, OLIG2 is also suggested to be of importance in this setting, driving relapse after irradiation treatment, specifically in *MYC* amplified MB [83]. However, in contrast to SOX9 leading to MYC downregulation, OLIG2 was linked to co-expression with MYC suggesting different roles in perhaps different states of the relapse process (Figure 2). Eventually, the Notch pathway has also been seen to play a role in Group 3 experimental models, promoting both metastasis and self-renewal [84].

### 3.4. Group 4 Medulloblastoma

Relapsed Group 4 MB differed most markedly from disease at diagnosis [54] and metastasis is a predictor of bad prognosis [55,85]. However, it has been shown that Group 4 MB relapses have a better overall survival than other non-WNT subgroups [24]. Perhaps related to this is that Group 4 MB with i17q or chr 11 loss have better survival than their Group 3 counterparts [55].

Enrichment of *CDK6* and *CDK14* co-amplifications can be seen in Group 4 tumor relapses. 17p and 11p loss were more common in Group 4 tumor relapses when compared to a reference cohort. When, instead, a cohort of paired primary-relapsed tumors were studied, relapses had losses of chromosomes 9p, 10p, 20p, and 20q. The mutations commonly found in Group 4 tumor relapses included *USH2A*, *DDX3X*, *CHD7*, *NEB*, *EPHA7*, and *GTF3C1* [54] (Figure 2).

### 3.5. Group 3 and Group 4 Medulloblastoma Subtypes

When comparing the transcriptome of matched primary:relapsed Group 3 and Group 4 MB samples, the expression of genes like PDIA6 and FKBP9 correlated with a poor prognosis and SNORD115-23 correlated with a better prognosis [27]. Further, deconvolution analysis of transcriptome data showed that Group 3 and Group 4 tumors presented with elevated cell cycle activity at relapse. When looking further at Group 3/4 subdivisions [86]; subtype II, characterized by *MYC* amplification, *GFI1*/*GFI1B* activation, *KBTBD4*, *SMARCA4*, *CTDNEP1*, *KMT2D* mutation, had a shorter time to relapse with an increase in distant disease; subtype VIII, characterized by PRDM6 activation, *KDM6A*, *ZMYM3*, *KMT2C* mutation, had longer time to relapse but also an increase in distant disease; subtype III, characterized by *MYC*/*MYCN* amplifications, also showed an increase in distant disease as compared to the time of diagnosis. Subtype V, characterized by *MYCN* amplification, and subtype VII, characterized by *KBTBD4* mutation, did not show an increase in distant disease. However, subtypes V and VII already had a higher incidence of spread at diagnosis; however, it is important to note that the number of cases is limited in this comparison [26]. Some evidence of a subtype switch within Group 3 and Group 4 tumors has been observed during relapse [54].

Time from relapse to death was significantly associated with the molecular features of relapse related to *MYCN* amplification and 3p loss (univariable) [54]. *TP53* mutation and intact 9q were even significant in a multivariable analysis across all subgroups (WNT, infant SHH, non-infant SHH, Group 3 and Group 4) [54]. Clinical data have highlighted MYC and p53 as important players in relapsed MB [79]. This p53 suppression is supported by our experimental models, where MYC can, via methylation of *ARF*, suppress the p53 pathway and that the loss of *ARF* in MYC-driven MB-like tumors increases the number of metastases [82].

Apart from overlaps in the different MB subgroups and subtypes described above (summarized in Figure 2), there are many molecular pathways of relapse suggested to be shared between relapsed MBs. BPIFB4 was seen to be upregulated in SHH and Group 4 relapses compared to primary disease, which also could be seen in in vivo models of relapse [87]. The BPIFB4 downstream target eNOS, could be targeted to treat recurrent Group 3 models, but, as with many of the other molecular characteristics of relapsed MB, it is mainly tied to cell self-renewal.

**Table 1 cancers-16-01752-t001:** A selection of MB mouse models for metastasis and relapse.

MB Model	Subgroup Resemblance	Model Type	Incidence	Latency (Months)	Metastasis/ LMD Rate	Refs.
mWnt-MB	Wnt	Orthotopic	100%	average of 1.7 m	0%	[57]
mWnt-MB, *Myc^T58A^*	Wnt	Orthotopic	100%	average of 0.5 m	80%	[57]

*Ptch1* ^+/−^	SHH	GEMM	28%–39%	6–8 m	50%	[68]
*Ptch1*^+/−^, Math1-cre; LSL- Atoh1	SHH	GEMM	100%	3–4 m	100%	[70]
*Ptch1*^+/−^, Atoh1-creER;LSL- Atoh1	SHH	GEMM	100%	6–7 m	100%	[70]
*Ptch1*^+/−^, Math1-SB11/T2Onc transposon	SHH	GEMM	97%	2.5 m	80%	[48]

RCAS *Shh*	SHH	Retroviral	40%	2–3 m	~10%	[71]
RCAS *Shh*, *Eras*; *Lhx1*; *Ccrk* or *Akt*	SHH	Retroviral	50–60%	1–2 m	30–45%	[71]

Heterzygous NeuroD2-*SmoA1*	SHH	GEMM	48%	6 m	0%	[73]
Homozygous NeuroD2-*SmoA1*	SHH	GEMM	94%	1–2 m	30%	[73]
NeuroD2-*SmoA1*, Math1-GFP	SHH	GEMM	100%	5.4 m	28%	[88]

Atoh1*-SmoM2*	SHH	GEMM	~50%	3–4 m	15–30%	[74]
Atoh1-*SmoM2*, *Kmt2d*^fl/+^ or *Kmt2d*^fl/fl^	SHH	GEMM	100%	2 m	65–100%	[74]

*Trp53mut*	SHH	GEMM	0%	-	0%	[48]
Trp53*mut*, Math1-SB11/T2Onc transposon	SHH	GEMM	40%	average of 3 m	100%	[48]

*MYCN* in iPSC-NES; hbNES	SHH	Orthotopic	90–100%	2–4; 4–6 m	30–70%; 0–30%	[64]
*MYCN* + *OCT4* in iPSC-NES; hbNES	SHH	Orthotopic	100%	1; 2 m	100%; 50%	[64]

GTML (Glt1-tTA, TRE-*MYCN*)	Group 3	GEMM	75%	2–6 m	~10%	[78]
GTML, *Trp53*^KI/KI^	Group 3	GEMM	100%	1.3–3.3 m	NA	[79]
GTS (MYCN/SOX9-driven)	Group 3	GEMM	100%	1.5–4 m after dox loss	50–60%	[80]

GMYC (Glt1-tTA, TRE-*MYC*)	Group 3	GEMM	62%	average of 4.3 m	~5%	[82]
GMYC, *ARF*^−/−^	Group 3	GEMM	95%	average of 3.3 m	50–60%	[82]

GEMM: genetically engineered mouse model; NA: not available, LMD: leptomeningeal dissemination, iPSC-NES: iPSC-derived neuroepithelial stem cells; hbNES: human hindbrain-derived neuroepithelial stem.

Leptomeningeal spread in MB has been associated with several molecular pathways. The chemokine receptor CXCR4 and its ligand CXCL12 (aka. SDF-1) promote cell migrations in granular precursors, the proposed cell of origin for SHH MB, and is also believed to play a role in tumor cell migration/metastatic spread [89]. The enzyme WIP1 (aka. PPM1D) is high in Group 3 and Group 4 MB, which also have high levels of CXCR4. WIP1 is not only a p53 regulator [90], it also inhibits the expression of GRK5, which phosphorylates CXCR4, leading to internalization of the receptor [91].

As previously mentioned, MB has been proposed to spread via the hematogenous route to form leptomeningeal dissemination, where experimental models of mainly SHH and Gr. 3 MB suggested this to be driven by CCL2/CCR2 signaling [34]. *CCL2* and *WIP1* are both located on chr 17q, with isochromosome 17q (i17q) being an important feature of aggressive MB [81]. Chromosome rearrangements like this likely serve as a single feature linked to several molecular pathways of malignancy.

## 4. Clinical Investigations with Implications for Relapsed/Metastatic Medulloblastoma

New therapies for pediatric malignancies, such as MB, are generally first tested in adults and then repurposed for the relevant cancers in a pediatric population due to safety. Compounds with inferior efficacy in adult cancers are therefore not evaluated in children. Additionally, trials are often conducted on a heavily pre-treated group of patients with a very poor prognosis. This is problematic, as new therapies are not tested in treatment-naïve patients, making it almost impossible to evaluate the potential of a drug or treatment regime to kill the relapse and metastasis-causing cells. This section is dedicated to reporting currently ongoing trials that are interesting from a high-risk MB perspective, in which MB patients with metastatic and relapsed disease are included, but not necessarily the main targeted patient population. Because of the rarity of these tumors and a tremendous need for new therapeutics in pediatric relapsing tumors, many studies are pooling multiple high-risk treatment-refractory/relapsed pediatric brain tumor patients with various diagnoses. Notably, very few clinical studies investigate treatments targeted specifically towards disseminated cancer cells/relapse-causing MB cells.

### 4.1. Targeting of SMO-Driven Medulloblastoma

Ten to fifteen years ago, several SHH inhibitors showed great results in preclinical models. Treating patients with SHH-driven MB looked feasible. SMO inhibitors such as sonidegib or vismodegib, efficiently bind to SMO and inhibit downstream SHH pathway activity, and showed good response, especially in patients with *PTCH*-mutated relapsed SHH tumors [92,93]. Still, SMO inhibitors had limited success and treatment resistance arose in tumors, driving mutations in downstream SHH pathway genes, including *GLI2* and *MYCN* [92,94]. Further, clinical trials with vismodegib led to severe side effects in children, including irreversible growth plate fusions [95]. The findings resulted in a trial amendment and restricted the use of this agent to skeletally mature patients. In recent research, promising new agents are being investigated to counteract resistance to SMO inhibitors such as small molecule inhibitors [96] or a sterol analog [97] acting on downstream targets to block pathway activation [98] or inhibiting SMO via alternative mechanisms [97]. In addition to targeting the SHH pathway itself, combination treatments blocking compensatory oncogenic signaling from other pathways such as MAPK and FGFR should be tested [99]. However, none of these approaches are currently being used in opened trials.

### 4.2. Targeting of MYC-Driven Medulloblastoma

Histone acetylation modifies chromatin organization and largely affects the expression and regulation of many oncogenes, including the *MYC* family of genes commonly amplified in MB. Histone deacetylases (HDACs) are enzymes that remove acetyl groups from histones. Bromodomain and Extraterminal domain (BET) proteins are readers that bind acetylated histones. The interplay of chromatin remodeling, *MYC* overexpression, and HDAC or BET inhibitors have suggested therapeutic strategies for MYC-driven MB [100,101,102]. In MYC-driven MBs, PI3K/AKT/mTOR signaling is also altered, with preclinical evidence that these inhibitors can target MB metastases and decrease growth in vitro and in vivo, with synergistic effects combined with BET inhibition [64,103] or HDAC inhibitors [104,105,106]. Clinical trials on BET inhibitors in children are hampered by dose-limiting toxicities related to the fact that inhibitors also target other bromodomain-containing cellular proteins. However, given encouraging preclinical data on HDAC and PI3K inhibition, PNOC016 (NCT03893487) was designed as both a target validation study (the primary objective) and a preliminary efficacy trial to evaluate drug penetration of the pan-HDAC and PI3K inhibitor fimepinostat for newly diagnosed diffuse intrinsic pontine gliomas, recurrent high-grade gliomas, and recurrent MB. Patients who are surgical candidates are eligible and receive the drug two days prior and on the day of surgery for the phase 0 PK/PD component of the trial and then continue the drug following surgical recovery. This study is presently closed, and results are pending.

The NCT04696029 Phase I trial of Eflornithine/DFMO is interesting as it includes molecular high-risk/very high-risk primary MB patients in its study population. The purpose of this trial is to evaluate relapse prevention by using Eflornithine as maintenance therapy for MB. Eflornithine is an old drug that has previously been studied in the context of neuroblastoma with some success [107,108]. Its target, ornithine decarboxylase (ODC), is a transcriptional target of MYCN and can be co-amplified with *MYCN* [107]. Interestingly, Eflorithine was shown to decrease MYC levels in pancreatic cancer [109]. Eflornithine/DFMO has not been studied in the context of MB and the high prevalence of *MYC*/*MYCN* amplification in high-risk MB, both primary and relapsed, which makes it intriguing to evaluate. As mentioned, it is currently unknown how and if Eflornithine/DFMO targets a poor-prognosis heterogenous MB tumor. However, if this inhibitor primarily targets the fast-dividing *MYC*/*MYCN* tumor cells and not the dormant cancer cells that cause the relapse, this study might not meet its endpoint. This study is presently recruiting in the US.

We and others have discussed the potential benefits of the CDK4/6 inhibitors palbociclib, ribociclib, and abemaciclib in the context of MYCN-driven MB and other brain tumors (reviewed in [110]). Since then, the number of clinical trials investigating this treatment avenue for pediatric brain tumors has exploded and is most commonly combined with chemotherapy and exclusively for Rb-positive tumors; some are currently open (e.g., NCT03434262, NCT05429502, NCT03387020, NCT03709680, NCT03526250, NCT03155620, NCT04238819). Preclinical evidence supporting CDK4/6 inhibition for MB is rather strong [111,112] and the understanding of regrowth mechanisms and cell states has expanded to spatial-transcriptomics resolution [113]. The fact that MB tumors become more homogenous by differentiating towards the neuronal lineage but with some potential for self-renewal implies that drug combination regimens are the way to go, as also suggested in [111].

Another relevant CDK-targeted therapy for high-risk/relapsed MB is CDK9 inhibition. Similar to BET-bromodomain inhibitors, CDK9 inhibition blocks MYC transcriptional output by interfering with transcriptional elongation. Fadraciclib is a dual CDK9/CDK2 inhibitor included in the ESMART basket trial (NCT02813135) [114], and this trial arm is currently recruiting. Concurrent action on this particular combination of CDKs is an auspicious approach, especially for primary or recurrent MYC-high MBs, as we previously discussed [110]. We have also reported successful combination treatment of CKD2 inhibition with BET-bromodomain blockade for primary MYC-driven MB [84], whereas others have targeted the transcriptional machinery with both CDK9 and BET-bromodomain inhibition [115]. It is too early to speculate how well these cell-cycle inhibitors will help our patients. The danger of this approach could present as an increase in the amount of tumor cells entering the previously discussed dormant, growth-arrested/slow-cycling state, protected against chemotherapy [112], thereby risking relapsed disease at a later time.

### 4.3. Immune-Therapies

Bevacizumab is an anti-VEGF monoclonal antibody that is used as an antiangiogenic therapy. For MB relapse, two different combinatorial treatment regimens have mainly been used and are tested with bevacizumab. First, the very recently published MEMMAT trial (NCT01356290) investigated metronomic multi-drug cytostatic treatment together with biweekly bevacizumab. This trial is currently recruiting for ependymoma and ATRT but is closed for MB relapse at the moment. This study, and earlier, preliminary studies have reported increased PFS and OS fhor patients with recurrent MB and showed potential for long-term survival [116,117,118]. The rationale behind this study was to use low dosages of multiple metronomically dosed cytostatic drugs in combination with antiangiogenic therapy to target the tumor-supporting microenvironment. This opens up the option of using drugs that the patients have been treated with previously and have even developed resistance against. With the limited options currently available for relapsed MB, this approach presents as feasible for a large proportion of patients. Second, bevacizumab has been tested in a three-drug regimen with temozolomide and irinotecan with promising results [119]. However, this study was reported to be inferior to the MEMMAT trial [117]. The major limitation of both of these trials is that they were not powered to conclude on the subgroup level.

A different approach to utilizing the potential of the immune system to target tumor cells is to activate T-cells cells through inhibition of the IDO pathway with indoximod. The exact mechanism of how indoximod exerts its effect is complex and not fully understood. It likely functions as a tryptophan mimetic, and partly blocks tryptophan catabolism, leading to the mTOR-mediated activation of T-cells. The effect of indoximod has been shown to improve when combined with DNA-damaging agents [120]. This approach is implemented in clinical trials for adult and pediatric brain cancers [121] (NCT02052648). For relapsed/refractory MB, indoximod is combined with temozolomide and radiotherapy (NCT04049669). The latter of these trials is still recruiting.

The FDA (US Food and Drug Administration) has given fast-track designation for the use of SurVaxM in newly diagnosed GBM. Survivin is a driver of mitosis and an inhibitor of apoptosis, among other things. It is highly expressed in multiple cancers, including MB. In MB, Survivin expression has been associated with poor prognosis and was proposed as a therapeutic target almost two decades ago [121,122]. Unfortunately, Survivin is difficult to target directly, and several alternative routes have been explored without much success. The SurVaxM/Montanide ISA 51 trial (NCT04978727) is another attempt in which a cancer vaccine is being used to train the patient’s immune system to recognize Survivin peptide presentation on cancer cells. Despite promising results for adult GBM patients [123] it is too early to speculate on the efficacy in a high-risk pediatric population. This trial is currently recruiting, with the primary objective being to evaluate Regimen Limiting Toxicity (RLT) or a Phase 1 trial.

While the majority of brain tumor clinical trials utilizing oncolytic virotherapy have been in adults, five viruses are being tested in pediatric brain tumors but none in randomized clinical trials. Most of the trials are currently closed: herpes simplex virus (G207) (NCT02457845), reovirus (pelareorep/Reolysin)(NCT02444546), measlesvirus (MV-NIS) (NCT02962167), poliovirus (PVSRIPO) (NCT03043391); but the adenovirus trial is currently recruiting in Spain (DNX-2401, AloCELYVIR) (NCT04758533).

Beyond oncolytic virotherapy, viruses can be used as vectors to deliver gene therapy. Apart from, e.g., adenoviruses and retroviruses that have been tested in childhood brain tumor trials [111,124], the use of less virulent adeno-associated viruses (AAVs) is proven to be safe and efficient [125] and opens new therapeutic possibilities.

Patient-engineered T-cells to form CAR-T can be directed toward relevant targets known to be expressed on MBs: HER2, B7-H3, EPHA2, GD2, PRAME207, TFs and GPC2 and Interleukin 13 receptor *α*2. HER2 and B7-H3 CAR-T cell trials are ongoing with catheter delivery: HER2-positive r-Pediatric CNS tumors (NCT03500991) [126], B7-H3 (NCT04185038) [127], while other target trials are under development at earlier stages. Dual CAR-T is also a promising development in the search to enhance a single CAR-T strategy or to reduce toxicity. From preclinical data on mouse models, Donovan et al. propose a locoregional delivery of novel immunotherapy by CAR T cells to the cerebrospinal fluid for the treatment of metastatic MB [128]. Intra-thecal delivery of CAR-T-cells against EPHA2, HER2, and IL13Rα2, a target identified by the Mitchell lab [129,130], is an effective treatment for primary, metastatic, and recurrent Group 3 MB and PFA ependymomas in mouse models [128].

## 5. Discussion and Future Ideas

The current clinical trials for primary metastatic MB include arms with intensified conventional therapy, including high-dose chemotherapy, with stem cell rescue and high-dose radiotherapy [14]. CSI is the standard radiotherapy for all medulloblastomas because of the high dissemination risk of this disease, and is given, in the case of HR MBs, in a higher dose to avoid disease relapse. These intensive therapies given to HR patients are toxic and the secondary effects of high-dose radiotherapy are detrimental to the young child, providing a limited quality of survival with severe cognitive impairment. Thus, alternative strategies are greatly needed to target the relapsing tumor cells, preferably already while treating the primary tumor.

Targeted therapies must be tested, and today, few phase II trials are open for children. Oncolytic viruses and CAR-T trials are opening in phase I, but precise delivery of these promising therapies is still to be solved as the intravenous route is not sufficient for efficacy in the CNS. Implantation of Ommaya reservoirs intratumorally or intraventricularly provides long-term access to CSF and will enhance drug delivery into the brain but is currently not the standard of care. In the MEMMAT protocol, intraventricular Ommaya reservoirs have been successfully used for treatment delivery at relapse and in targeted therapy trials [126,128,131].

To understand tumor progression and relapse mechanisms/biology in detail, studies on paired samples of primary:relapse MB biopsies with molecularly comprehensive subgroups would be useful. Up to now, relapses have not been shown to share many common mutations or frequently defined chromosomal rearrangements that can be targeted or inhibited. This suggests that one general relapse-specific treatment will not be enough. Rather, several different individualized therapies are likely necessary, addressing a cell state phenotype evolution in the relapse tumor compared to the primary tumor. Alternatively, profiling DNA, mRNA, and proteins at a single-cell resolution will likely provide novel information about the clonal evolution of therapy-resistant cells in paired primary:relapse samples.

More studies are needed to link published data on specific proteins upregulated in metastasis and relapsed MB to the mechanisms ruling these cell states. If LMD is the cause of a more aggressive and recurrent disease for non-WNT MBs, targeting drivers of LMD is a promising strategy to inhibit the development of relapse and metastasis. Still, this would likely only be truly beneficial for MB disease without metastasis at diagnosis. Instead, targeting already developed LMD or relapses, perhaps even the MET cell state (if it exists), would be a compelling alternative. Preferably, apart from novel treatment strategies, earlier and more reliable detection of metastatic disease must be implemented. If the altered cell states involved in promoting LMD or metastasis maintenance can be identified, such states could be suppressed, as suggested in glioblastoma [132]. Putting efforts into this challenge could reveal promising targets of the growth-arrested tumor cells that can escape conventional chemotherapeutics as well as the immune system, already at primary disease.

Studying MB, but also other pediatric brain tumor entities with a high-LMD prevalence at diagnosis, could be useful in finding novel targeted strategies. Here, the GEMMs that develop LMD that we described (Table 1), or PDX models from paired primary:relapse MBs that disseminate during tumor progression, can be useful to model relapse mechanisms in better detail. Although mouse models are useful tools for the functional evaluation of genes involved in driving LMD or relapse, they either need to be immunodeficient to facilitate PDX establishment or rely on aberrant mouse (rather than human) cancer genes in order to model tumorigenesis in vivo with an operating immune system. The use of mice engineered with a functional human immune system might be a better alternative to mimic human disease, albeit expensive. Alternatively, 3D models like tumor organoids or assembloids, where tumor cells can be grown successfully in organoids, could be studied or used in drug screening. In addition, large-scale efforts, such as genome-wide or smaller targeted CRISPR/Cas9 screens anchored with a baseline standard regimen or experimental drug at inferior doses in vivo, could be useful tools to characterize drivers or relapse.

Finally, not enough is known to have warranted targeted therapies against dormant and metastasis/recurrence-causing tumor cells in MB patients. Such treatments could, in theory, complement chemotherapeutics or targeted drugs towards oncogenic drivers, killing tumor cells over the whole spectrum, from dormant to proliferating. An example of this is the notion that dormant cells activate autophagy to survive in their non-dividing state [133]. Targeting autophagy in breast cancer as a means of stopping recurrence caused by dormant tumor cells is currently being investigated in patients (NCT03032406, NCT04523857, NCT04841148). As the field is expanding and perhaps as the general genetic or epigenetic mechanisms behind dormancy are better understood, this might also change the way in which patients with high-risk MB and MB relapses are treated. Importantly, targeted therapies against dormant MB cells could then be integrated into primary treatment, eradicating therapy-resistant cells before clinical relapse is observed. Hopefully, given that dormant cancer cells are more radioresistant, some of these therapeutics could further allow for a decrease of the radiation doses given to the developing child.

## 6. Conclusions

The most common malignant pediatric brain tumor, medulloblastoma, has recently been intensively profiled molecularly. There are numerous mouse models generated trying to resemble this tumor entity, that comprise four different subgroups, often so diverse that they could be considered different cancer types. All research efforts dedicated to categorizing medulloblastoma have profoundly advanced our understanding of this developmental disease.

Despite this knowledge, MB relapse is often unpredicted and fatal. It is difficult to foresee, and the genetic or epigenetic changes driving relapse are often unknown and likely different, regardless of subgroup. It is still known that leptomeningeal spread increases with tumor progression and often accumulates in MB relapse.

In this review, we summarized research reports describing MB metastasis and relapse and identified MB metastasis and relapse models that could be used to test novel treatments targeting therapy-escaping cells. We have given several examples of the genetic changes found in relapsed patient samples and identified novel strategies used to gain more biological knowledge about dissemination within the CNS. Finally, we highlighted clinical trials that are open and therapeutic strategies that are used to combat MB relapse. We hope this review can be used as an important resource in the MB research field and encourage further studies on brain tumor metastases and relapse, to find novel treatments that could advance outcomes of this disease.

## Figures and Tables

**Figure 2 cancers-16-01752-f002:**
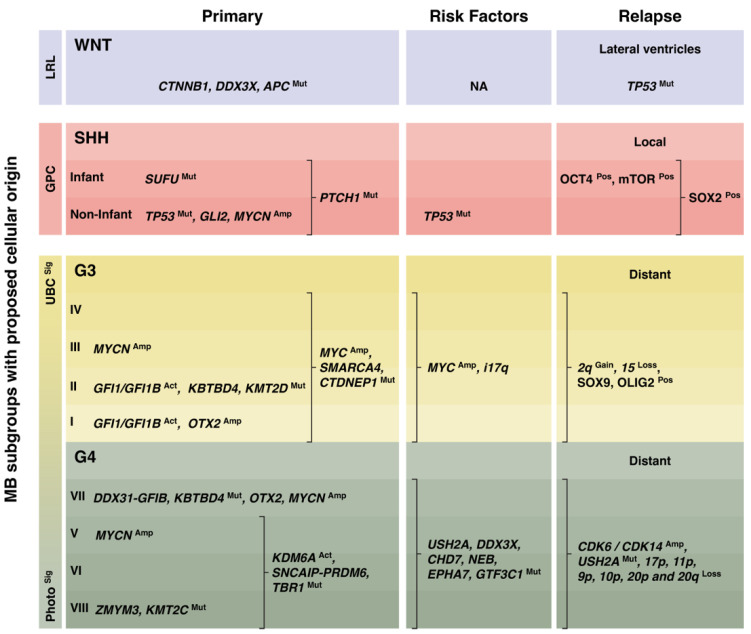
Potential molecular changes driving medulloblastoma progression and relapse. Primary medulloblastoma (MB) subgroups with highlighted molecular drivers and proposed cells of origin are shown [21]. Risk factors state molecular markers associated with relapse. Molecular markers listed in the relapse column have been described in tumors samples at relapse. The last column also states localization of relapse dependent on subgroup. WNT: WNT-activated, SHH: Sonic hedgehog activated, G3: non-WNT/non-SHH Group 3. G4: non-WNT/non-SHH Group 4. LRL: lower rhombic lip, GPC granular precursor cell, Photo^Sig^: Photoreceptor signature, UBC^Sig^: unipolar brush cell signature. Mut: mutated, Amp: amplified, Act: activated, Pos: positive.

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
