# Peer review of "Drivers Underlying Metastasis and Relapse in Medulloblastoma and Targeting Strategies"

_cancers, 2024, doi:10.3390/cancers16091752_

Round 1
Reviewer 1 Report
Comments and Suggestions for Authors
Holmberg et al. provide a thorough review of the biological and clinical patterns associated with recurrent and metastatic medulloblastoma. The article is well written and a useful resource that addresses a critical issue in advancing medulloblastoma outcomes.
I have just a couple of suggestions that could strengthen this review:
1) SHH with p53 mutation and Gp3 with MYC amplification are known to have the highest rates of recurrence of all medulloblastoma variants. What is known about SHH with p53 mutation and Gp3 with MYC amplification specifically with respect to metastasis at presentation and 1st relapse?
2) For completeness of this review, it would be helpful to provide more information on those studies that have compared the biology of primary site tumor to metastasis and/or presentation to recurrence in human samples. Similar to table 1, an up-to-date list of these studies, such as Wang et al. Acta Neuropath, 2015 and Okonechnikov et al. Acta Neuropath Comm, 2023, would strengthen this paper.
Author Response
We thank the reviewer for taking time to read this review and for the encouraging words that this can be an important resource in the MB field and in studying metastases and relapse. Please find a point-by-point response to the concerns addressed below.
- Thanks for this comment, we agree that it would be good to include details on this and have added additional explanatory text regarding these patient subtypes in the section “Metastatic disease and relapse patterns” on page 2. Perhaps this is best studied in Cavalli data set from Cancer Cell 2017 or from the review from Hovestadt et al. NatCanRes 2020. We have now included some text to point out that p53 mutations correlate with SHH alpha subgroup and have high metastasis rates at diagnosis (approx 20%). Still, the SHH beta subgroup has even more metastasis at diagnosis (33%) and still fewer cases with p53 mutations.
-
Thanks for the comments of a suggested up-to-date list of such comparative studies from human samples. We have included and referred to several papers in the review already explaining differences in mutations or methylation of samples. However, we are happy to also include papers comparing transcriptomic data as suggested. Please find descriptions of central findings from these two reports and an inclusion of their references on page 4 in the introduction section of “Molecular signatures of medulloblastoma relapse” but also on page 6 (subsection: SHH medulloblastoma) and on page 8 (subsection: Group 3 and Group 4 medulloblastoma subtypes).
Wang et al showed that molecular subgroup is stable between primary and metastatic disease by studying a matched sets of primary-metastatic biopsies using expression and methylation profiling. This work suggested that the MB subgroups arise from distinct cells of origin but that the primary and metastatic compartments of these MBs are sharing a distinct cell of origin.
More recently Okonechikov et al studied the transcriptome of MB and showed the impact of age in where the changes in gene expression profiles were more pronounced in the younger SHH-MB patients with relapses. For Group 3 and Group 4 MB samples, expression of genes like PDIA6 and FKBP9 correlated with poor prognosis and SNORD115-23 correlated with better prognosis. Further, deconvolution analysis of bulk transcriptome data showed that SHH tumors became more dedifferentiated and Group 3 and Group 4 tumors presented with elevated cell cycle activity at relapse.
Reviewer 2 Report
Comments and Suggestions for Authors
Overall, this review offers a thorough examination of metastasis and relapse in MB. Nevertheless, there are some aspects that could be improved to enhance the article.
1. What is the frequency of IR-induced 2nd tumors, particularly brain tumors, in survivors of medulloblastoma compared to survivors of other childhood IR-cancers?
2. In Fig 2 and line 223, the author briefly mentions that mTOR signaling promotes MB relapse and metastasis, but it would be more comprehensive to also consider the role of AMPK pathway.
3. In the section discussing the "targeting of SMO-driven MB" (line 398), it is crucial for the author to highlight and provide updates on recent findings and clinical trials involving the use of SMO inhibitors. Additionally, updating the potential of combining these inhibitors with other drugs to minimize drug toxicity in pediatric MB patients is also important.
Comments on the Quality of English Languagen/a
Author Response
Thanks. Please find a point-by-point response to the concerns addressed below:
- Thank you for this comment. Yes, cranial irradiation is associated with a higher incidence of developing a secondary brain tumor in pediatric patients but not adults. We now included the following information on page 5. Here we describe that: About 9% of long-term survivors of primary CNS cancers will have developed a second CNS cancer 40 years later (PMID: 35814854). Irradiated pediatric MB patients have a higher risk to develop a secondary brain tumor (mostly meningiomas but also glioblastomas) as compared to other brain tumors excluding pediatric patients with glioblastoma.
- Thanks. We now included this section after discussing mTOR signaling:
“AMP-activated kinase AMPK inactivation also slows the progression of SHH-driven medulloblastoma. Disabling AMPK reduces mTORC1 activity and impairs MB stem cell (OLIG2-positive) populations (PMID: 38094249). AMPK can further suppress metastasis of SHH-MB cells by restraining the activation of SHH and NF-κB pathways where dual blockade of these pathways have a synergically therapeutic effect on SHH MB (PMID: 36683064).”
- We thank the reviewer for these suggestions and have updated this section. We have now included text about alternative targeting of the SHH pathway (SMO and downstream target GLI) as well as suggested combination treatments with MAPK and/or FGFR inhibitors (PMID: 26130651) MAPK inhibitors are currently being evaluated in BRAF-driven pediatric low-grade brain tumors. It would therefore not be a major leap to test these in MB.
Reviewer 3 Report
Comments and Suggestions for Authors
This is a review paper on drivers underlying metastasis and relapse in medulloblastoma and targeting strategies.
The review jumps forth and back with biological and clinical information. I suggest to first stay with biological information and to proceed with clinics later on. E.g., the authors introduce the SIOPE HR trial already in the first chapter (but do nor mention the European HR trial which has been recently concluded).
Page 2, line 88 ff. „About 5-7% of all MBs relapse as 88 High-Grade Gliomas [26,27].“ I do not think this is true. It’s rather sequence of tumors that share a genetic background, or are based on irradiation. and not as much a relapse from medulloblastoma. Plus clarify or add additional, convincing literature.
The part on target mechanisms and drugs is well written and does not need much improvement.
In summary, the paper needs significant restructuring and a clear order of topics as incidence, epidemiology, biology, mechanisms of metastasis, clinical trials, a.s.o. It also would profit from a more scientific language in some parts.
Author Response
- Thanks for this advice on improving the overall structure of this review. The idea was to give a brief introduction about the clinical information, then move over to biology and finish off with treatments and clinical trials. It feels a bit logical to us to not jump into biology before we describe the clinical nature of the disease and the relapse mechanisms involved. That said we now tried to structure the review better to make it more focused. Hope it all looks better. Also, we have thoroughly reviewed the literature with the help of the SIOPE working group for medulloblastoma and looked for earlier high risk medulloblastoma trials and found that SIOPE HRMB was the first trial of this kind. For a European trial we also looked for a trial that included more than 3 countries. And the current open trial is based on national trials, from Germany, France or UK, the latest published in 2016
- Thank you for giving us the chance to clarify our view on this. As stated in the text, we also believe that these secondary malignancies are not true MB recurrences but rather treatment induced secondary cancers or unrelated development of a glioma. We have changed the phrasing of this section to make our point clearer. We added a few more sentences and literature in this in this section and also included a section where we discuss different radiation treatment modalities.
- Thanks, we have ordered this a bit better now. In introduction, MB incidence and epidemiology is now included to give the paper a better start. Then we move into metastasis at diagnosis and recurrence in the different MB subgroups. We also tried to remove lay language and included more scientific language.
Reviewer 4 Report
Comments and Suggestions for Authors
Thank you for the opportunity to review this paper on molecular mechanisms underlying medulloblastoma progression and relapse.
The title of the review effectively captures the main focus of the paper, which is to summarize the molecular mechanisms driving medulloblastoma progression and relapse, discuss preclinical models, and highlight clinical trials and therapeutic strategies. The introduction provides helpful background on medulloblastoma diagnosis, risk stratification, and the prevalence of metastasis at diagnosis vs relapse. Figures showing the relapse frequency by subgroup and a diagram of the molecular changes driving progression are useful/informative. The authors effectively integrate insights from clinical data, preclinical models, and ongoing trials to present a clear picture of the state of the field.
The review is well-structured, with helpful visual aids and a particularly useful table summarizing mouse models of metastasis and relapse for different MB subgroups. The authors thoroughly discuss the biological processes enabling leptomeningeal dissemination, the molecular markers associated with relapse in each subgroup, and the means of detecting and targeting minimal residual disease. The section on clinical trials is informative, highlighting relevant studies of SMO, MYC, and CDK inhibitors as well as various immunotherapies. However, the discussion could benefit from a more critical analysis of the pros and cons of each therapeutic strategy. Similarly, the conclusion would be strengthened by additional insights into future research directions and the most promising therapeutic avenues.
Overall, this review tackles a topic of high importance given the dismal outcomes for relapsed MB. The authors have done an excellent job of thoroughly reviewing a large body of literature and providing a valuable resource for the field. With minor revisions to incorporate more critical perspectives and future outlooks, this manuscript merits acceptance for publication.
Thank you to the authors for their significant contribution to our understanding of medulloblastoma biology and treatment. This comprehensive review will undoubtedly serve as a key reference for researchers and clinicians working to improve outcomes for patients with this challenging disease.
Author Response
We thank the reviewer for the kind words and helpful feedback regarding the analysis of the current treatments and trials as well as improving the conclusion. We have now included more critical analysis on different treatment options in the discussion. Additionally, we have expanded the conclusion with our thoughts on the most promising treatments as well as our view on where research resources should be allocated to best combat MB relapse.
Reviewer 5 Report
Comments and Suggestions for Authors
Review
The manuscript comprehensively reviews the molecular mechanisms driving the progression and relapse of medulloblastoma, which is a highly heterogeneous cerebellar tumour. The scope of the article includes the biology of tumour spread, insights from preclinical models for studying tumour recurrence, and goes on to outline ongoing clinical trials aimed at treating cancer cells causing relapse. By combining insights from biological studies with clinical trial data the authors propose novel therapeutic strategies targeting the mechanisms of metastasis and relapse. However, the article could be significantly improved if the authors provide their interpretation of the findings of the cited studies, rather than simply describe them. See ‘detailed comments’ for specific examples.
Completeness of review:
The review thoroughly covers the biological aspects of medulloblastoma progression, and highlights key molecular pathways involved in tumour progression and relapse. The authors identify gaps in understanding, in particular the role of dormancy and the mesenchymal-to-epithelial transition in lepto-meningeal metastasis.
Relevance of the review:
The subject area is highly relevant as medulloblastoma relapse has a grim prognosis. The review’s focus on molecular drivers and potential therapeutic targets is important for the future development of more effective and less toxic treatments.
Cited references:
These all seem appropriate on the whole. However, references should be cited at the end of the first sentence describing the corresponding study/studies, and not several sentences later which is sometimes the case.
Figures and table:
The figures succinctly convey to the reader the points that the authors make in the text. For Figure 1, most of the smaller rectangles lack a sharp point. There should be a better explanation of the abbreviations in the legends of Fig. 2 in particular, e.g. LRL, etc. The table omits a key early GEMM model of SHH-MB harbouring Ptch1/Tp53 compound mutations (Wetmore et al, CANCER RESEARCH 61, 513–516, January 15, 2001). In addition the table should be improved with explanation of the abbreviations, e.g. T2Onc2, GTML, etc.
Detailed comments:
1. Line 158 – state upfront which types of MB would be amenable to detection by ctDNA. SHH-MB is typically located laterally in the cerebellar hemispheres.
2. Line 203-206 – the in vivo and in vitro findings on ALCAM appear to be divergent. The authors should speculate briefly about why that might be.
3. Line 242 - why does deletion of Atoh1 prevent SmoM2-induced medulloblastoma? Is there disruption of the GCP lineage or a lineage switch that prevents SmoM2 oncogenic activity? Line 244-6: Related to this, why does Atoh1 overexpression in Ptch1 mutant mice enhance SHH-MB development? We know Atoh1 is connected to ciliogenesis in granule precursors (Chang Dev Cell 2019, DOI: 10.1016/j.devcel.2018.12.017) so perhaps there an indirect effect on the efficiency of SHH transduction? Another possibility is that ATOH1 overexpression expands the pool of granule precursors thereby increasing the probability of oncogenic transformation initiated by Ptch1 loss-of-function?
4. Line 305-6: provide a reference for the first sentence of this paragraph.
5. Line 324 -25. This sentence is unclear and should be re-phrased.
6. Line 333-34. “USH2A has been associated with…” What is the significance of this/why does the reader need to know this?
7. Line 541-545 should be rephrased as it is hard to understand.
Comments on the Quality of English Language
In general this is good, but there are a few sentences that could be better structured.
Author Response
Thanks for careful evaluation of our review. We agree with the comment that we should provide more interpretations of the findings of the cited studies, rather than simply describing them. Further, we provide a point-by-point response to deal with the other concerns here.
Regarding cited references:
Thanks for noticing this. We have now move some references up earlier, but we will discuss with editors in detail how/where they want to have these presented.
Regarding Figures and Table:
In order to explain the Figure 1 a bit better. The rectangles shown lack a sharp point by purpose. The fade is to illustrate that there is variation in recurrence time point, especially since we here are showing an overview based on the 4 main subgroups. Hope this makes sense.
Thanks for also pointing out the nice study by Wetmore et al. from 2001. When we selected the reference studies in our Table, we only focus on the studies with reported metastasis or leptomeningeal dissemination. In this study, the Loss of p53 dramatically accelerated the age of onset and the incidence of medulloblastoma in Ptc+/- mice - but authors do not mention anything about metastasis or leptomeningeal dissemination. That is why we did not select this study in our review.
“In addition the table should be improved with explanation of the abbreviations, e.g. T2Onc2, GTML, etc.” Thanks for this suggestion. We agree with reviewer, we added more explanation of the abbreviations in table and figure legends.
Regarding detailed comments:
1. We have now clarified this better on page 4. In recent publications the availability of ctDNA in CSF is from multiple medulloblastoma subtypes (PMID: 33110059).
2. Thanks for pointing this out. We now added this sentence in the text “These contradictory results between ALCAM function in vitro and in vivo indicates that ALCAM may be affected by the surrounding microenvironment.”
3. Yes, I guess reviewer means Line 342-Line 346. We agree with reviewer, we should interpretate the findings of the cited studies rather than simply describe them. We added our interpretation in the text.
“We speculate that granule neuron progenitors (GNPs), the tumor origin of SHH MB, proliferate by transducing Sonic Hedgehog (SHH) signaling via the primary cilium, and Atoh1 controls ciliogenesis and GNP pool expansion, thereby increasing the possibility of tumor initiation, perhaps metastasis.”
4. We have now added the reference below to support our claim that MYC amplifications are at least 3–4 times as common as MYCN amplifications in Group 3 MB. Northcott et al states that 17% of Gr 3 MBs have MYC amplification, while only 5% have MYCN amp. (PMID: 28726821).
5. Thank you for pointing this out. We have rephrased this sentence that now says: “However, it has been suggested that Group 4 MB relapses have a better overall survival than other subgroups”. We hope that this change is satisfactory for the reviewer.
6. We agree with this concern. Findings of USH2A in Group 4 MB were likely only correlative and we thus removed this sentence.
7. We agree with the reviewer and have rephrased the sentence accordingly. The new text reads:
“Implantation of Ommaya reservoirs intratumorally or intraventricularly is currently not standard of care. In the MEMMAT protocol, intraventricular Ommaya reservoirs have been successfully used for treatment delivery at relapse and in targeted therapy trials [126, 128, 131].”
Round 2
Reviewer 5 Report
Comments and Suggestions for Authors
The fluency of the narrative should have been improved before re-submission.
Comments on the Quality of English LanguageSub-optimal.